# South Asian black carbon is threatening the water sustainability of the Asian Water Tower

Junhua Yang [1], Shichang Kang [1,2] ✉, Deliang Chen [3], Lin Zhao[4], Zhenming Ji [5], Keqin Duan[6], Haijun Deng [7], Lekhendra Tripathee [1], Wentao Du [1], Mukesh Rai [1], Fangping Yan[1], Yuan Li [1] & Robert R. Gillies [8,9] ✉

Long-range transport of black carbon from South Asia to the Tibetan plateau and its deposition on glaciers directly enhances glacier melt. Here we find South Asian black carbon also has an indirect effect on the plateau's glaciers shrinkage by acting to reduce the water supply over the southern Tibetan plateau. Black carbon enhances vertical convection and cloud condensation, which results in water vapor depletion over the Indian subcontinent that is the main moisture flux source for the southern Tibetan plateau. Increasing concentrations of black carbon causes a decrease in summer precipitation over the southern Tibetan plateau, resulting in 11.0% glacier deficit mass balance on average from 2007 to 2016; this loss rises to 22.1% in the Himalayas. The direct (accelerated melt) and indirect (mass supply decrease) effects of black carbon are driving the glacial mass decline of the so-called "Asian Water Tower".

The Tibetan plateau, also referred to as "Asian Water Tower", is a critical water resource for Asia[1,2], affecting the health and welfare of billions of people who depend upon the headwaters of the Tibetan plateau[3,4] to feed the rivers of their region. Since a huge quantity of water flows out from the Tibetan plateau, water vapor supplied from outside the plateau essentially maintains the Tibetan plateau's water balance[5,6]. The external water vapor sources for the Tibetan plateau originate primarily from the Arabian Sea and Bay of Bengal−transported by the South Asian monsoon[4,5,7]; these sources regulate the precipitation variability over the southern Tibetan plateau[7,8]. Consequently, precipitation trends and variability over the southern Tibetan plateau could serve as indicators to predict changes in the sustainability of the Tibetan plateau's water supply. Moreover, precipitation supplies mass for the glaciers; precipitation has been predominantly

declining since the 1990s, resulting in an acceleration of glacial shrinkage across the Tibetan plateau over the past 30 years[2,9]. This glacial shrinkage ultimately affects the "reservoir" function of the plateau and, in turn, future water resources for Asia[9].

Summer (June–September) precipitation accounts for more than 60% of the total annual precipitation[5,10,11] over the Tibetan plateau, and its spatiotemporal variations are complex[1]. Changes in summer precipitation have tended to exhibit regional trends−i.e., increasing over the north while decreasing over the south of the Tibetan plateau[6,12]. Declines in summer precipitation supply inevitably results in glacial shrinkage over the southern Tibetan plateau; this is especially prevalent in the Himalayas[2,13]. Meanwhile, located in the upstream aspect of moisture transported to the Tibetan plateau, South Asia has experienced an uptick in summer monsoon precipitation since 2002[14].

[1]State Key Laboratory of Cryospheric Sciences, Northwest Institute of Eco-Environment and Resources, Chinese Academy of Sciences, Lanzhou 730000, China. [2]University of Chinese Academy of Sciences, Beijing 100049, China. [3]Regional Climate Group, Department of Earth Sciences, University of Gothenburg, Gothenburg 40530, Sweden. [4]Key Laboratory of Land Surface Process and Climate Change in Cold and Arid Regions, Chinese Academy of Sciences, Lanzhou 730000, China. [5]School of Atmospheric Sciences, Key Laboratory of Tropical Atmosphere-Ocean System, Ministry of Education, Sun Yat-sen University, Zhuhai 519082, China. [6]School of Geography and Tourism, Shaanxi Normal University, Xi'an 710119, China. [7]School of Geographical Sciences, Fujian Normal University, Fuzhou 350007, China. [8]Utah Climate Center, Utah State University, Logan, UT, USA. [9]Department of Plants, Soils, and Climate, Utah State University, Logan, UT, USA. ✉e-mail: shichang.kang@lzb.ac.cn; robert.gillies@usu.edu

South Asia is recognized as a region with particularly acute air pollution when compared with the rest of the world[15]. Many studies have emphasized the effect of aerosols on the South Asian monsoon precipitation regime[16–21], principally owing to the radiation absorption potential of black carbon. While the aforementioned studies indicate differing consequences of black carbon impacts, they all remark upon the fact that the presence of black carbon in the atmosphere generates additional warming in the South Asian monsoon region; this induces tropospheric instability and moisture convergence at lower levels and subsequently triggers convection and precipitation. However, it remains unclear (a) to what extent South Asian black carbon has influenced long-range moisture transport to the Tibetan plateau; (b) whether South Asian black carbon has affected the observed precipitation decrease over the southern Tibetan plateau; and (c) the overall influence of black carbon concerning the water sustainability of the plateau.

The analysis described in this paper provides evidence that South Asian black carbon conceivably reduces precipitation over the southern Tibetan plateau, which in turn diminishes the water supply of the Tibetan plateau, especially in the Himalayan region. Concomitant to the precipitation loss is the glacial mass decline; such a circumstance has substantial implications for the water balance of the so-called "Asian Water Tower", and for the subsequent delivery of an adequate water supply to downstream regions in the future. This study is based on an integrative analysis, including in-situ observations, reanalysis datasets, numerical simulations, and statistical analyses.

## Results

### Precipitation trends over the Tibetan plateau and South Asia

The southern Tibetan plateau experiences the maximum intensity of summer precipitation as compared to the entire plateau as shown in Fig. 1a; this reflects the incoming water vapor source to the plateau, i.e., it is closely associated with the South Asian monsoon and local convection over the Indian subcontinent. Although precipitation over the Tibetan plateau has overall increased over the past half century[22,23], it exhibits singular spatial distributions, especially since the 1980s. Observations indicate an increasing precipitation trend over the northern Tibetan plateau with a reversal in trend, albeit slightly negative, over the southern Tibetan plateau[6]. Based upon our analysis of monthly gridded precipitation data from the Climatic Research Unit (CRU, University of East Anglia, Norwich, UK), we calculated the accumulative anomaly in summer precipitation and determined that there had been an increased summer precipitation trend over the southern Tibetan plateau, but this trend was reversed in 2004 (Fig. 1b). Moreover, the Global Precipitation Climatology Project (GPCP) monthly precipitation dataset (Supplementary Fig. 15) corroborates what is shown in Fig. 1b. Further detail reveals that a slight increase in summer precipitation emerged over the past 56 years for both the southern (Fig. 1c) and northern Tibetan plateau (Supplementary Fig. 15). However, as noted previously, beginning in 2004, the southern Tibetan plateau summer precipitation trend reversed, with a marked decrease of 4.4 mm/a (Fig. 1c). In addition, daily precipitation data from 86 meteorological stations confirm the existence of a decreasing trend for the southern Tibetan plateau during 2004–2016 (Supplementary Fig. 16a); this is accompanied by an increasing trend for the northern Tibetan plateau.

We subsequently calculated the 2004–2016 summer precipitation variation trend over the southern Tibetan plateau and adjacent regions, as noted in Supplementary Fig. 16b. A decreasing trend in summer precipitation is evident over the southern Tibetan plateau, as compared to a somewhat noteworthy increasing trend over the Indian subcontinent for the most part, except for certain regions within central India. The aforementioned trends are, in turn, echoed by the inflexion point that was observed to have occurred in 2004 (Fig. 1b).

To further elucidate, a modicum increase in summer precipitation has occurred over the past 56 years in both South Asia and the southern Tibetan plateau. However, breaking it down into periodic components from 1961 to 2003, summer precipitation significantly decreased by 1.4 mm per annum (mm/a) in South Asia but insignificantly increased in the southern Tibetan plateau (statistical regression shown in Fig. 1c). When commencing in the year 2004, the summer precipitation trend changed, with a very distinctive increase of 9.9 mm/a in South Asia alongside a decrease of 4.4 mm/a in the southern Tibetan plateau. These results were also confirmed by the Asian Precipitation–Highly Resolved Observational Data Integration Towards Evaluation of Water Resources (APHRODITE)[24] data (Fig. 1c).

The key inquiry at this juncture, given the shifts in summer precipitation over the southern Tibetan plateau and in South Asia since 2004, calls for a process-oriented evaluation of long-range moisture transport and precipitation deposition in South Asia up to the southern Tibetan plateau.

### Linking black carbon to Tibetan plateau precipitation

In the southern Tibetan plateau, the impact of localized surface evaporation is not particularly pronounced[8,25]. Thus, the inter-annual variability of the summer precipitation regime is controlled primarily by the "long-range" transport of moisture. Figure 2a shows the monthly specific humidity and wind vectors from the ERA-Interim reanalysis during the summer months for the period 1989–2018. The meridional averaged (80°–90° E) moisture and circulation fields verify summer moisture transport from South Asia to the Tibetan plateau. However, of particular note, is the change in meridional moisture content and the vertical wind components between the periods 1989–2003 and 2004–2018 (Fig. 2b): since 2004, the southern Tibetan plateau slopes have experienced a reduction in specific humidity—opposite is the case over South Asia. The wind fields suggest that the "long-range" transport of moisture may have been reduced as evidenced by decreasing southerly flow wind speeds and enhanced upward motions. Hence, we calculated the vertical integral of moisture flux and moisture flux divergence for the period 2004–2018, and compared it with that for the period 1989–2003. The comparison (Supplementary Fig. 17) confirms a decrease in incoming moisture over the southern Tibetan Plateau but a strengthening of moisture convergence over South Asia. Further details regarding the moisture budget and moisture convergence are given in Supplementary Section 2.2.

Previous studies have emphasized the role that black carbon can play in inducing atmospheric moisture convergent[21] conditions and enhancing local convection in South Asian monsoon regions[19]. Therefore, we charted the black carbon emission trends in South Asia (Fig. 1b), which show an ever-increasing trend that began in 1985. Utilizing the CRU precipitation and Peking University's emissions datasets, we established that from 2004 onwards, the summer precipitation regime over the southern Tibetan plateau exhibited a significant negative correlation with the summer black carbon emissions variability over South Asia (Fig. 2c); this is in contrast to an insignificant positive correlation from 1961 to 2014 (Fig. 2d). Moreover, the statistically significant areas of Fig. 2c agree with those areas of increased summer precipitation indicated by Supplementary Fig. 16b. These results suggest that increased South Asian black carbon (Supplementary Fig. 19) has played an important role in the decrease of summer precipitation over the southern Tibetan plateau since the 21st century. In addition, to identify if there was a delayed effect of increased South Asian black carbon on reduced summer precipitation over the southern Tibetan Plateau, we conducted a lag correlation analysis and, found no lag relationship between the two (see Supplementary Section 4.1).

To further examine the cause and effect of atmospheric black carbon over South Asia on summer precipitation over the southern

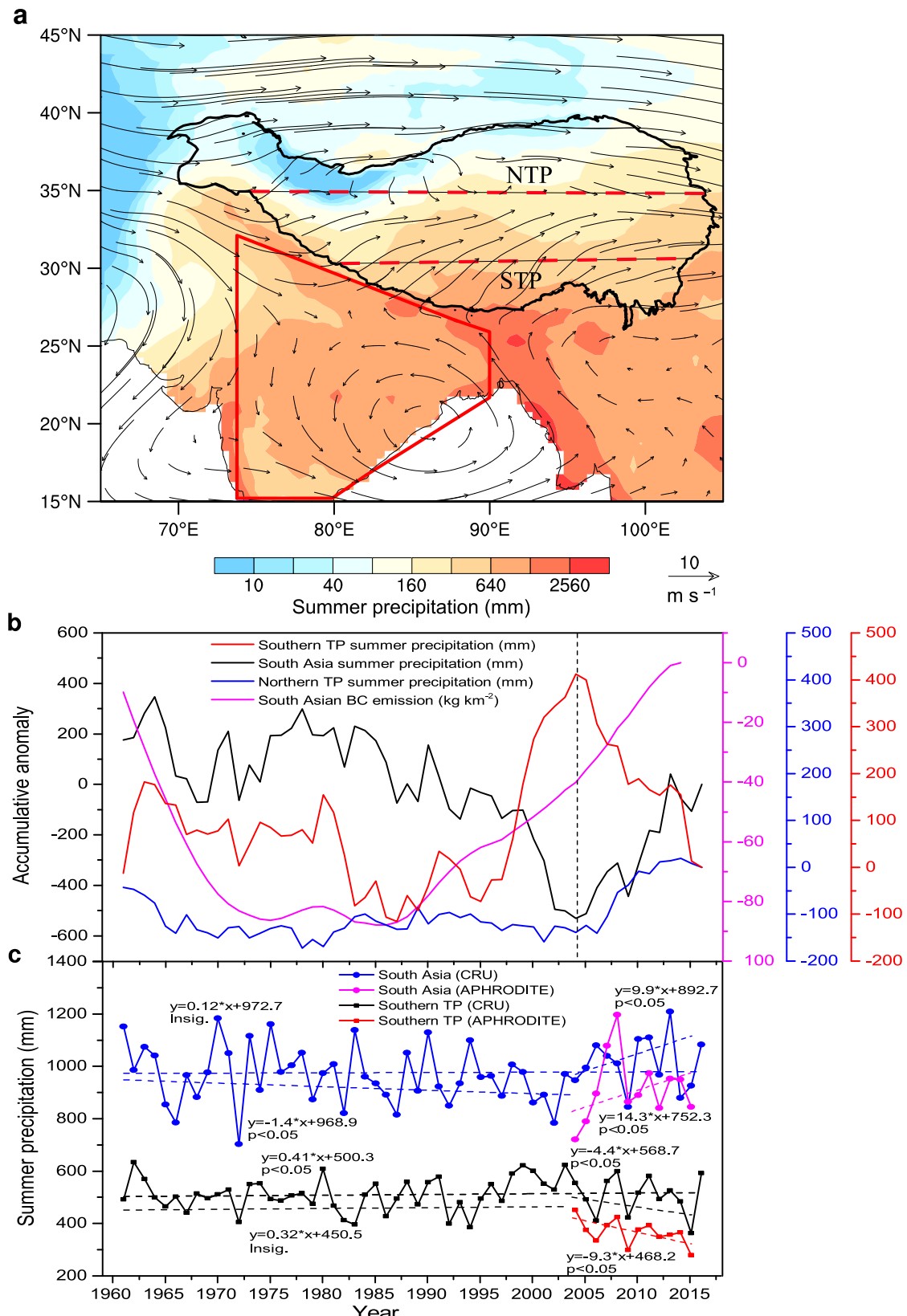

Tibetan plateau, a series of sensitivity simulations were conducted using the Weather Research and Forecasting model with Chemistry (WRF-Chem, see "Methods" for details) for summer months for ten years (2007–2016) after 2004. As noted in Fig. 3a, South Asian black carbon resulted in an increase of summer precipitation of up to 200 mm over South Asia (especially in eastern India), but a decrease of

approximately 100 mm over the southern Tibetan plateau. Moreover, the spatial distribution of summer precipitation changes is in general agreement with the spatial variation observed in the CRU data analysis (supplementary Fig. 16b). Both provide complementary evidence in support of the linkage between recent diminishing summer precipitation over the southern Tibetan plateau with increased black

**Fig. 1 | Summer precipitation and black carbon emission characteristics.**
**a** The spatial pattern of summer precipitation using the Climatic Research Unit (CRU) dataset overlain with summer 500 hPa wind fields for the period 2001–2016, using the ERA-Interim dataset over the Tibetan plateau (TP) and South Asia (denoted by red polygon). NTP and STP delineate the northern and southern Tibetan plateau boundaries. **b** Plots of the accumulative anomaly in summer area-averaged precipitation over the southern Tibetan plateau, the northern Tibetan plateau, and South Asia using the CRU dataset (1961–2016) along with, South Asian

area-averaged black carbon (BC) emissions using Peking University's emissions dataset (1961–2014). **c** Plots of observed summer area-averaged precipitation over the southern Tibetan plateau and South Asia using the CRU (1961–2016) and Asian Precipitation–Highly Resolved Observational Data Integration Towards Evaluation of Water Resources (APHRODITE, 2004–2015) datasets. Dashed lines represent linear trends broken into three separate periods (1961–2016, 1961–2003, and 2004–2016). Trends are statistically significant (*p*-value <0.05) unless otherwise stated.

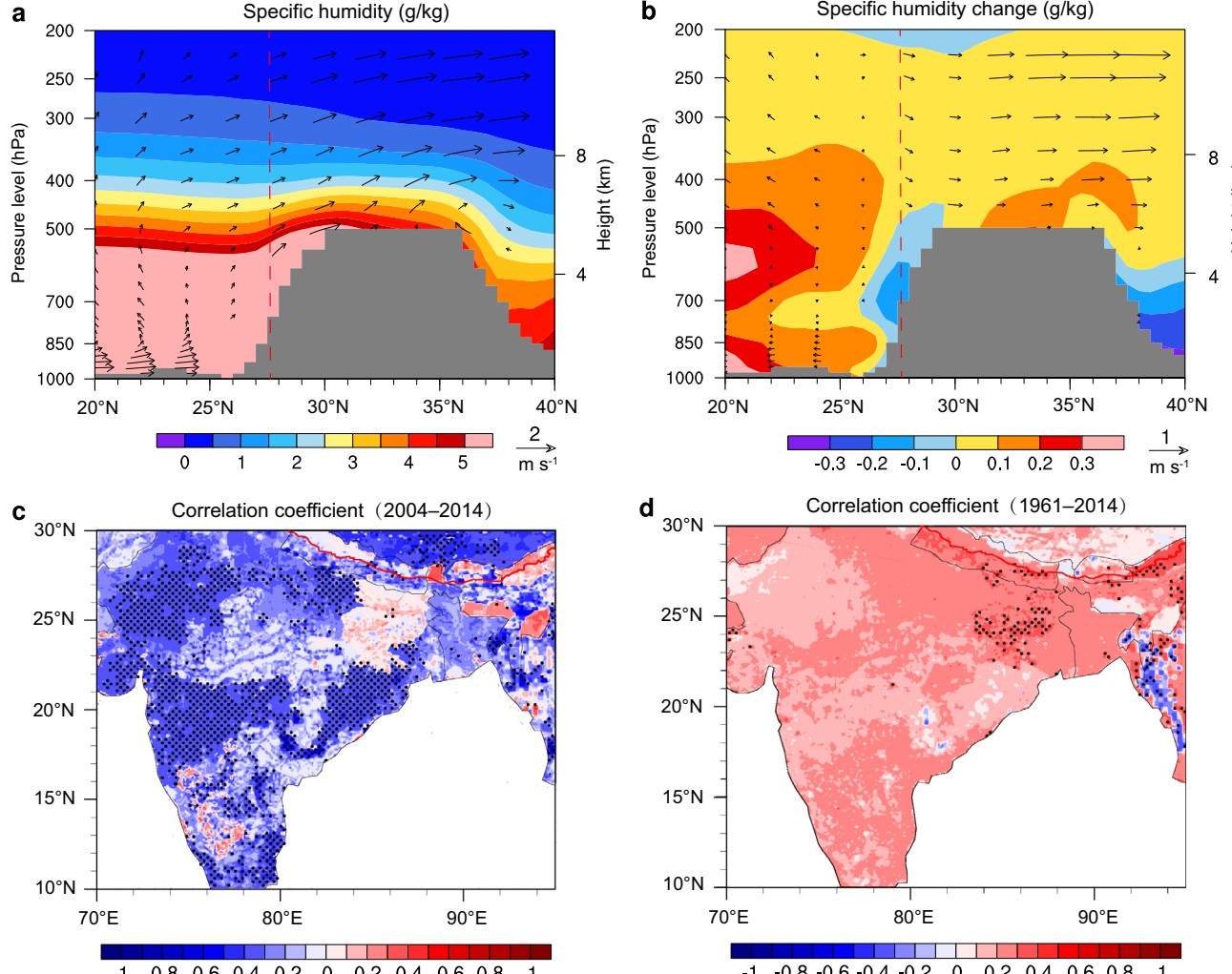

**Fig. 2 | Meridional moisture transport (using the ERA-Interim dataset) from South Asia to the Tibetan plateau, together with the linkage of South Asian black carbon to summer precipitation over the southern Tibetan plateau.**
**a** Cross-section of meridional averaged (80°–90° E) moisture transport for the period 1989–2018. **b** Cross-section of meridional averaged moisture transport change between the periods 1989–2003 and 2004–2018. **c, d** Correlation coefficients between area-averaged precipitation over the southern Tibetan plateau using a subset of the CRU dataset (1961–2014) to concur with Peking University's

black carbon emissions dataset (1961–2014) over South Asia. Black dots in **c** and **d** indicate statistically significant areas. Insignificant positive correlations in **c** still imply a decrease in summer precipitation over the southern Tibetan plateau from 2004 onwards is closely related to increased South Asian black carbon (Supplementary Fig. 19). In panels **a** and **b**, the vertical red dashed line represents the southerly extent of the Tibetan plateau while the gray area outlines the areal extent of the plateau.

carbon emissions originating from South Asia. In addition, South Asian black carbon plays a minor role in the decline of summer convective precipitation over the southern Tibetan plateau as indicated in Fig. 3b, despite the fact that convective precipitation is primarily controlled by local surface evaporation and low-pressure development. In contrast, the modeled large-scale precipitation (Fig. 3c) indicates a more marked precipitation reduction; this implies that South Asian black carbon emissions appreciably hinder the long-distance transport of water vapor onto the Tibetan plateau.

Next, the summer moisture flux change due to South Asian black carbon was investigated. The following aspects of change were noted, as shown in Fig. 4a. Incoming moisture from the southern boundary of the Tibetan plateau was reduced, coupled with a strengthening of the cyclonic circulation of moisture in the eastern Indian subcontinent and Bay of Bengal in both the low- (Supplementary Fig. 24) and middle- (Fig. 4a) tropospheric layers. The change in dynamics brought about an enhancement of the vertical transport of moisture, which decreased the water vapor in the low troposphere (Supplementary Fig. 24), while

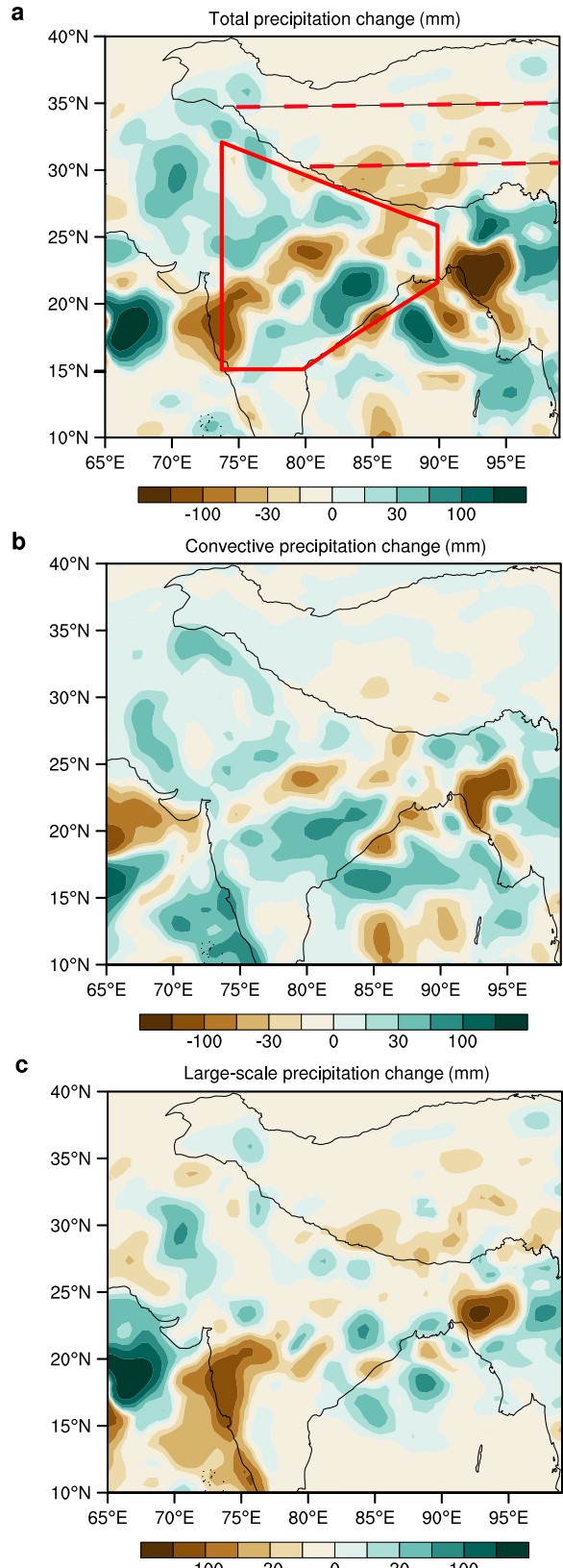

**Fig. 3 | Summer precipitation change simulations using the Weather Research and Forecasting model with Chemistry (WRF-Chem). a** Averaged summer precipitation change due to South Asian black carbon for the period 2007–2016; this is then broken down into two components in **b** convective precipitation and **c** large-scale precipitation. Precipitation change estimates were derived based upon WRF-Chem simulations, where precipitation output from the control simulations (initialized with South Asian black carbon) was compared with that from the sensitivity simulation (devoid of South Asian black carbon). South Asia is delineated by the solid red polygon and the Tibetan plateau is outlined by red dashed lines depicting its northern and southern regions, as noted in Fig. 1.

temperatures and their changes caused by black carbon in the troposphere (Supplementary Fig. 26), and found an increase in the meridional temperature gradients in the eastern Indian subcontinent, i.e., northern-increased versus southern-decreased temperatures. The enhanced meridional temperature gradient tends to cause the intensification of zonal wind vertical shear according to thermal wind balance considerations and enhanced cyclonic circulation[26,27]. Further analysis of convective available potential energy (CAPE) confirms a significant increase in CAPE indicative of increased convection in the eastern Indian subcontinent (Fig. 4c).

In addition to the direct radiative effect, black carbon can also influence liquid and ice cloud properties[28–31]. As documented in Fig. 4d, summer weighted vertically averaged cloud condensation nuclei (CCN) number concentrations, at 1% supersaturation, were markedly increased over the Indian subcontinent. As previous studies have suggested, an increase in CCN tends to suppress precipitation[32], especially in the case of light rain[33]. However, other studies have found that increasing the number of anthropogenic CCN leads to enhanced precipitation[34,35], whereby cloud particle size increases with increasing aerosol loading over moist regions with high water vapor content and strong convection conditions[29].

There are abundant water vapor sources in South Asia and the region is host to a rich convective environment[36–38]. As shown in Supplementary Fig. 27, South Asian black carbon results in higher summer cloud water mixing ratios over the Indian subcontinent, which can facilitate localized precipitation. When enhanced localized precipitation occurs as a result of CNN forcing and convective enhancement, water vapor availability downstream (in the direction of the southern Tibetan plateau) is marginalized.

An additional event analysis was also undertaken. Two hundred and twenty-eight heavy rain days over the Indian subcontinent were selected from the daily gridded precipitation data of APHRODITE that ends at 2015, which accounted for 39.6% of total summer precipitation during the period 2007–2015. A heavy rain day[39] was defined as a day when maximum precipitation exceeded 100 mm. Somewhat surprisingly, daily heavy rain averaged over South Asia always accompanies comparatively low values over the southern Tibetan plateau (Supplementary Fig. 28). The WRF-Chem model is suitably configured and applicable to examine heavy precipitation events (see Supplementary Section 1.3 for details of the model evaluation). Simulated results indicated that black carbon lead to increased daily precipitation in most parts of the Indian subcontinent during heavy rain days (Supplementary Fig. 29a), and its presence is consistent with the location of the maximum precipitation centers (black dots in Supplementary Fig. 29a) as obtained from the APHRODITE dataset. Moreover, enhanced convective precipitation may well explain a large part of the increased total precipitation over the Indian subcontinent during heavy rain days (Supplementary Fig. 29b), while the decrease in large-scale precipitation (Supplementary Fig. 29c) explained most of the reduced precipitation over the southern Tibetan plateau. Therefore, the event analysis reinforces our earlier investigation of black carbon's role in enhancing convective instability and increasing summer heavy rain.

increasing it in the middle troposphere (Fig. 4a). At the same time, the northward transport of moisture to the southern Tibetan plateau was weakened (Fig. 4a); this was accompanied by enhanced moisture convergence in the eastern Indian subcontinent but increased moisture divergence on the southern Tibetan plateau (Fig. 4b).

The strengthening of the cyclonic circulation is due to the fact that black carbon heats the troposphere. We compared the air

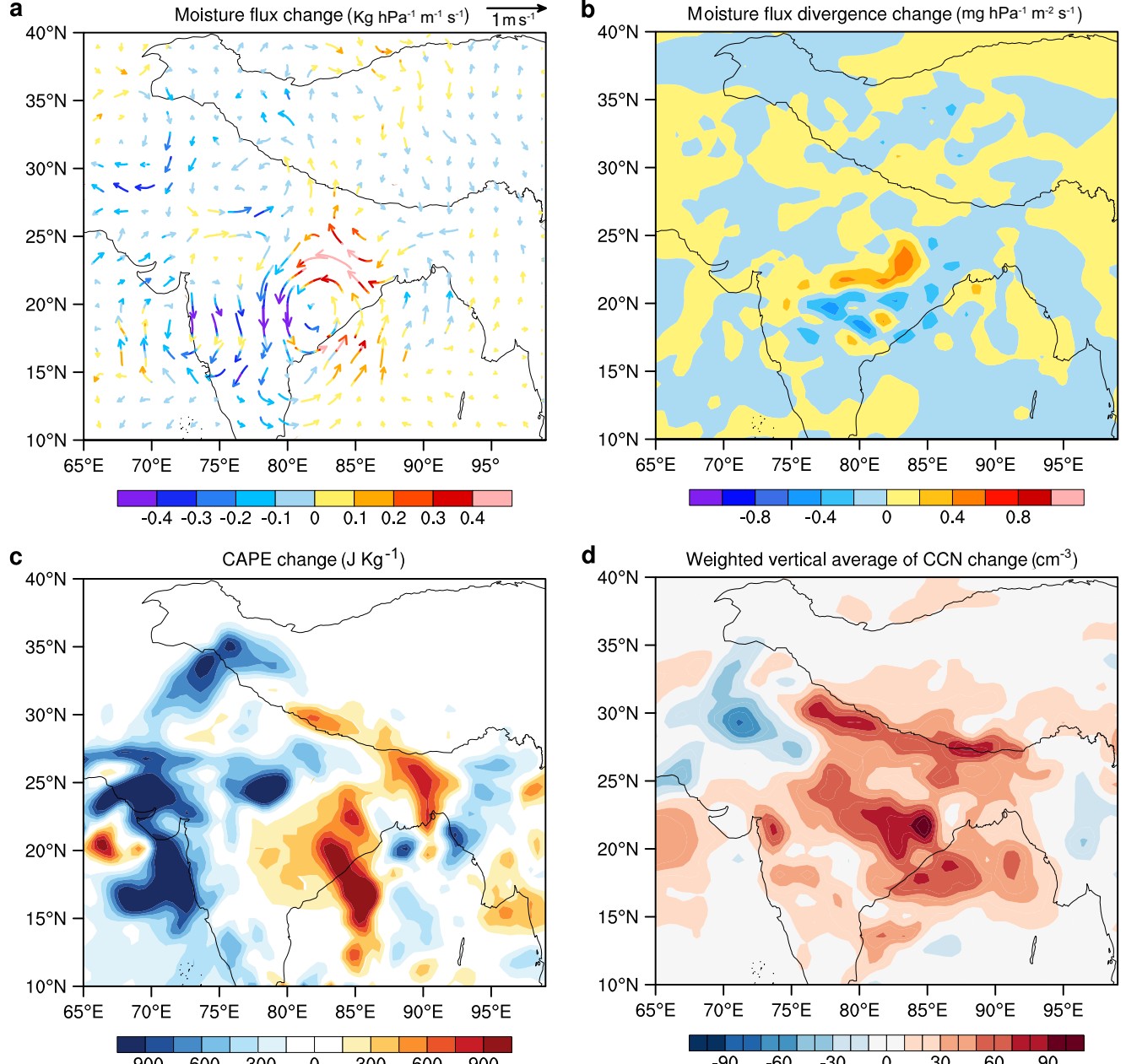

**Fig. 4 | Summer meteorological conditions change simulations using the Weather Research and Forecasting model with Chemistry (WRF-Chem).** For the period 2007–2016: **a** South Asian black carbon triggered average summer moisture flux change. **b** Averaged moisture divergence change at 500 hPa. **c** Averaged convective available potential energy (CAPE) change. **d** Weighted vertical average number concentration of cloud condensation nuclei (CCN) change. Changes in the aforementioned variables, as a result of South Asian black carbon, were calculated based on WRF-Chem simulation output; i.e., the values of the variables in the control simulations (with the effects of South Asian black carbon) minus those of the sensitivity simulations (without the effects of South Asian black carbon) as noted in the text.

**Further aspects for the water balance of the Tibetan plateau**

Summer precipitation decline over the southern Tibetan plateau signifies a risk for disturbing the water balance over the Tibetan plateau, given that advected moisture across the southern boundary is essential to maintain the water balance. Through a series of numerical simulations by the WRF-Chem model, it was discovered that increasing black carbon originating primarily in South Asia, has resulted in an increase in localized summer precipitation there and a reduction in the summer precipitation over the southern Tibetan plateau. Such a change in precipitation regimes will undoubtedly affect the Tibetan plateau's water supply. The simulated precipitation change was used to quantify the effect of South Asian black carbon on the Tibetan plateau's water supply; this was accomplished by using the Integrated Valuation of Ecosystem Services and Trade-offs model (InVEST). The results, given in Fig. 5, show that South Asian black carbon has brought about an overall decrease in the Tibetan plateau's water supply with the exception of some northern parts; such atmospheric modifications will disrupt the equilibrium water budget of the region. Of greater consequence, however, is the considerable reduction in the southern and central Tibetan plateau's water supply with the largest decrease in the Himalayas (up to 200 mm/annum). It is not by coincidence that these stretches have experienced recent glacial retreat, especially the Himalayas[13]. The result of reduction in the water supply is inevitable negative glacier

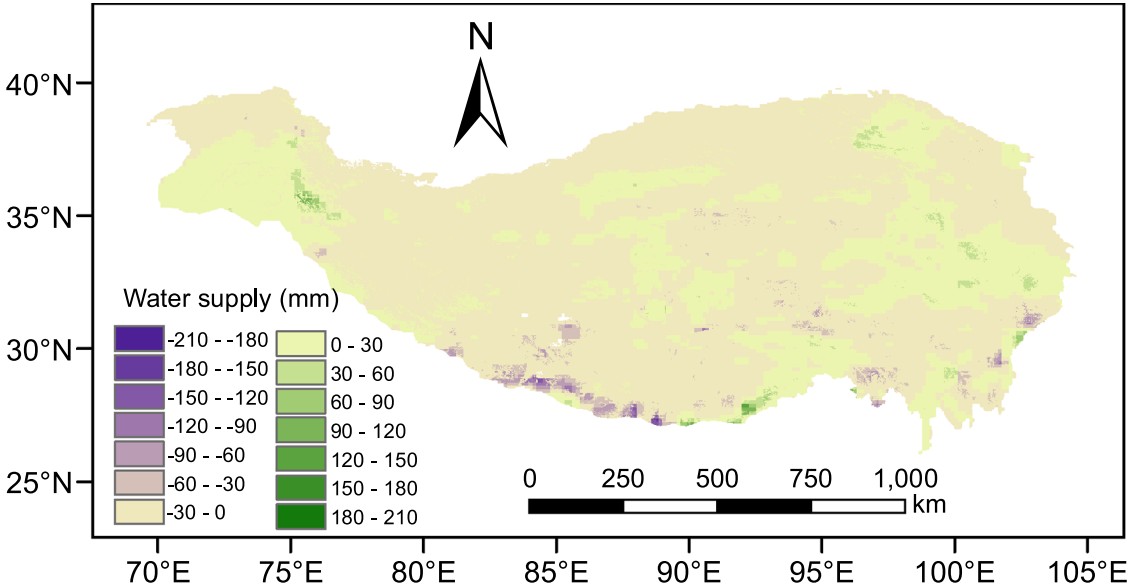

**Fig. 5 | South Asian black carbon-initiated averaged summer water supply change over the Tibetan plateau for the period 2007–2016.** The base map is generated based on the calculation using the Integrated Valuation of Ecosystem Services and Tradeoffs model.

mass balance, albeit that atmospheric warming is often ascribed as the dominant factor causing glacier shrinkage.

To expound upon the extent of glacial mass decline, the method proposed by Brun[40] was applied, where time series of digital elevation models (DEMs) were used to calculate the glacier volume changes over the Tibetan plateau for the period 2007–2016 period (see Supplementary Section 5.1 for further details). As outlined in Supplementary Fig. 30, the largest glacier mass loss appears in the southern Tibetan plateau with variability of −0.53 m water-equivalent-year⁻¹. Moreover, through the application of a monthly-scaled glacial mass balance model (Supplementary Section 1.4), we also found that there was a larger cumulative glacier mass loss in the southern Tibetan plateau (e.g., central Himalayas and Nyainqêntanglha for the period 1979–2014; Supplementary Fig. 31). The decrease of water (mass) supply caused by South Asian black carbon accounted for 11.0% of glacier deficit mass balance over the southern Tibetan plateau from 2007 to 2016; this rises to 22.1% in the Himalayas. As further evidence, we ascertained the drivers of glacial mass balance change using the monthly-scaled glacial mass balance model (Supplementary Section 5.2). The change in glacier mass balance fluctuates with precipitation perturbations (Supplementary Fig. 32b). Summer precipitation, a key contributor to glacier mass accumulation, has decreased during the period 1979–2014 in the central Himalayas, southern Tibetan Plateau (Supplementary Fig. 34b). Regardless of attribution, any reduction in glacial mass will threaten the sustainability of water resources over the Asian Water Tower and affect the water supply for those countries that rely upon it.

## Discussion

Since the beginning of the 21st century, South Asian black carbon emissions have been pivotal in altering summer precipitation over the southern Tibetan plateau; such emissions have reduced long-range moisture advection and subsequent precipitation regimes to and within the southern Tibetan plateau. Existing modeling studies concerning black carbon have detailed its impact on the South Asian summer monsoon[19,41,42], but the conclusions have been inconsistent. For example, Meehl et al.[41] suggested that black carbon aerosols reduced monsoon precipitation over India during the period 1988–1999, while Soni et al.[42] indicated that black carbon resulted in increased precipitation over northern India but decreased precipitation over southern India. These contradictory conclusions could be

caused by the temporal variation of black carbon emissions over South Asia, which increase from 1961 to 2014 (see Supplementary Fig. 19). The difference in South Asia's black carbon emissions compared with the global average has become larger since 21st century; this might have induced significantly higher radiative perturbations than those implied by the globally mean estimates[17].

In this study, a combination of lag-time correlation analysis and WRF-Chem simulations confirmed that South Asian black carbon emissions may have reached a sufficient level to significantly affect local convection; this is in agreement with the results of previous studies[16,17]. Through modeling, we ascertained that black carbon's indirect effect was to significantly increase CCN number concentrations over the Indian subcontinent. Moreover, black carbon's direct radiative effect enhances the vertical transport of moisture, which provides favorable conditions for stronger convection. Through the combination of the direct and indirect climate effects, the observed buildup of South Asian black carbon in the atmosphere can augment localized convection and precipitation and, reduce the moisture content available for advection downwind; the subsequent repercussion being a reduction in summer precipitation over the southern Tibetan plateau.

The integrative analysis presented here highlights the effects of historically increasing black carbon emissions that originate from the heavily industrialized and densely populated regions of South Asia; if such emissions persist, they will continue to reduce the southern Tibetan plateau summer precipitation. Furthermore, a probability density function analysis showed that there is a decadal shift in precipitation over the southern Tibetan Plateau (Supplementary Fig. 35b), causing a decadal shift in glacier mass accumulation (Supplementary Fig. 35c). Our analysis indicated that lower summer precipitation since the beginning of the 21st century, caused by South Asian black carbon, has reduced the mass gain for the glaciers over the southern Tibetan plateau. What is more, South Asian black carbon long-range transport to the Tibetan plateau and deposition on the glaciers enhances glacier melt in the region by 15% while, at the same time shortening snow cover duration[43–45]. The combination of direct deposition of black carbon, resulting in reduced albedo, decreased summer precipitation and higher air temperatures will, without a doubt, intensify glacial deterioration which consequently will alter the water balance of the Tibetan plateau; in turn, this has serious implications for the human population of the Tibetan plateau and surrounding regions. Therefore, mitigating South Asian black carbon emissions would be judicious if

one aims to maintain the water balance of the Tibetan plateau and so, avoid possible future water supply scarcities as well as geohazards such as glacial lake outflow floods that, to date, have been devastating in certain countries in South Asia.

## Methods

### Study domain and core data

The study domain consists of the Tibetan plateau and South Asia. As noted in Fig. 1a, the Tibetan plateau was divided into the northern Tibetan plateau (NTP), southern Tibetan plateau (STP), and a transition region between the two, following Yao et al.[2].

Observed daily precipitation data were obtained for 86 national observational stations from the China Meteorology Data Service Centre (CMDSC) to analyze precipitation trends for the period 1961–2016 over the study domain. The CRU dataset was used to calculate the temporal trend of precipitation variation within the study domain; this is a long-term global-scale monthly gridded precipitation dataset at a spatial resolution of 0.5°. In addition to the CRU dataset, daily gridded precipitation from the APHRODITE dataset, at 0.25° × 0.25° resolution, was utilized; this dataset provides a long-term continental-scale precipitation perspective over the study domain and has been applied previously in the study of precipitation variation characteristics over the Tibetan plateau and South Asia. Monthly moisture and wind vectors were derived using the ERA-Interim reanalysis dataset, with a gridded resolution of 0.25°; thus dataset was acquired from the European Centre for Medium-Range Weather Forecasts (ECMWF)[46]. Monthly global black carbon emission inventories for human activities were broken down by sector (energy production, industry, transport, residential and commercial, agriculture, and deforestation and wildfire), at 0.1° × 0.1° resolution; this inventories were obtained from Peking University[47].

### WRF-Chem model simulations

The WRF-Chem model is a newly developed regional dynamic/chemical transport model which simulates gas-phase chemical and aerosol microphysical processes along with numerous meteorological fields. WRF-Chem simulations were conducted at 25 km horizontal resolution, covering the study domain, with 190 grid cells in the east-west direction and 160 in the north-south direction. The detailed model setup is listed in Supplementary Table 1. The initial and boundary conditions for the meteorological fields utilized the 6-hourly National Center for Environmental Prediction final analysis data (NCEP FNL), at a horizontal resolution of 1° × 1°. The default application of anthropogenic emissions was obtained from the Intercontinental Chemical Transport Experiment Phase B (INTEX-B), including CO, $SO_2$, $NO_X$, VOC, BC, OC, $PM_{2.5}$, and $PM_{10}$, which were replaced by the real-time output from MOZART[48] (Model for Ozone and Related chemical Tracers) at 6-hourly intervals in this study. In addition, initial chemical conditions were updated by the real-time MOZART results. Biogenic emissions were obtained from the Model of Emission of Gases and Aerosol from Nature (MEGAN)[49] at a monthly time resolution. Fire emissions were obtained from the Fire INventory from NCAR (FINN)[50] at an hourly resolution. Further details of the spatial and temporal resolutions of these emission datasets are given in Supplementary Table 2.

The selection of physical and chemical schemes was based on the performance of different WRF-Chem model configurations (Supplementary Table 3). In Supplementary Section 1.3, we fully evaluated the WRF-Chem performance for precipitation (Supplementary Figs. 1–5 and Table 4) and black carbon (Supplementary Figs. 6–11 and Tables 5–6) based on in-situ observations and reanalysis datasets. The model reproduced satisfactory results for summer black carbon concentration and precipitation as well as their spatial variations and magnitudes. The model configuration when using the Kain-Fristch cumulus scheme[51], CBMZ gas-phase chemical mechanism[52], and MOSAIC aerosol module[53] had the most effective simulation performance, although low bias of black carbon concentration occurred at sites with complex terrain (Supplementary Fig. 11). Meanwhile, we also analyzed the model uncertainties caused by different spatial resolutions and cumulus schemes (Supplementary Figs. 12–14).

To investigate the impact of South Asian black carbon on the summer precipitation over the southern Tibetan plateau, two groups of WRF-Chem simulations were carried out. In the control simulations, the initial concentrations and emissions for black carbon were unchanged, including biogenic emissions, fire emissions, and anthropogenic emissions. Whereas in the sensitivity simulation, the initial concentrations and all the emissions of black carbon were set to zero over South Asia (the red polygon in Fig. 1a) at each time interval. The difference between the control and sensitivity simulations was thus used to indicate the effects of South Asian black carbon. The simulations began in May and ended in September for each year during the period 2007–2016; the month of May in each year was used as the model spin-up time.

### Water supply estimation

We used the InVEST (3.7.0) water yield model to estimate the impact of South Asian black carbon on the water supply of the Tibetan plateau. The water supply $Y(x)$ for each grid on the landscape $x$ was determined as the precipitation minus the fraction of water that undergoes evapotranspiration as follows:

$$Y(x) = \left(1 - \frac{AET(x)}{P(x)}\right) \cdot P(x) \tag{1}$$

where $AET(x)$ is the actual evapotranspiration for grid $x$ and $P(x)$ is the precipitation on grid $x$. The evapotranspiration portion of the water balance, $AET(x)/P(x)$, is based upon an expression of the Budyko curve proposed by Fu[54] and Zhang et al.[55]:

$$\frac{AET(x)}{P(x)} = 1 + \frac{PET(x)}{P(x)} - \left[1 + \left(\frac{PET(x)}{P(x)}\right)^{\omega}\right]^{1/\omega} \tag{2}$$

where $PET(x)$ is the potential evapotranspiration, provided by the Global Potential Evapo-transpiration (Global-PET)[56] dataset, which is based on modeling and analyses by Antonio Trabucco at 30 arc seconds (~1 km at the equator). $\omega$ at grid point $x$ is a non-physical parameter that characterizes the natural climatic-soil properties, determined by plant available water content and root depth.

### Glacier mass balance model

According to the method proposed by Radić and Hock[57], the glacier mass balance for mountain ranges over the Tibetan Plateau during the period 1979–2014 was accomplished by using a monthly-scale mass balance model. Glacier area-weighted specific mass balance ($B$) for the whole glacier in each mountain range was calculated as a sum of the specific mass balance ($b$) of each elevation band on a glacier ($i$):

$$B = \frac{\sum_{i=1}^{n} b_i \cdot S_i}{\sum_{i=1}^{n} S_i} \tag{3}$$

where $b_i$ and $S_i$ denote specific mass balance and glacier area, respectively, and the subscript ($i$) represents the number of the elevation band on the glacier ($i = 1, 2, 3, ..., n$) with an elevation interval of 50 m.

The monthly specific mass balance ($b_i$, mm water equivalent) of each elevation band on a glacier was calculated as:

$$b_i = a_i + c_i + R_i \tag{4}$$

where $a_i$ is glacier surface ablation (negative), $c_i$ is glacier mass accumulation (positive), and $R_i$ refers to snowmelt refreezing (positive) at each elevation band. Further details for the calculation of $a_i$, $c_i$, and $R_i$ are introduced in Supplementary Section 1.4.

## Statistical analysis

We estimated the precipitation variation trends using least-squares linear trend analysis and by applying the Mann-Kendall trend statistical test. The correlation between precipitation and black carbon over the Tibetan plateau and South Asia was determined using Pearson's correlation coefficients and the two-tailed Student's t-statistic. Accumulative anomaly[58] was used to reveal the turning year in the changes of precipitation and black carbon emissions. The detailed formulas are introduced in Supplementary Section 1.1.

## Data availability

CRU gridded precipitation data used in this study are available for download at https://crudata.uea.ac.uk/cru/data/hrg/. APHRODITE precipitation is available at https://www.chikyu.ac.jp/precip/english/downloads.html. The ERA-Interim reanalysis dataset is accessed at https://cds.climate.copernicus.eu/cdsapp#!/search?type=dataset. Global black carbon emission inventories are available from Peking University, at http://inventory.pku.edu.cn/. NCEP FNL data are accessible at https://rda.ucar.edu/datasets/ds083.2/. MEGAN code and data can be found at https://www.acom.ucar.edu/wrf-chem/download.shtml. FINN data are available from NCAR, at https://www.acom.ucar.edu/Data/fire/. The Global-PET dataset is available at https://cgiarcsi.community/data/global-aridity-and-pet-database/. Observations of precipitation and black carbon for stations cited in the text are available at http://shichang-kang.sklcs.ac.cn/data-sharing.html. MOZART results are no longer available online as of March 18, 2022; however, those used in this study are available upon request from Shichang Kang (shichang.kang@lzb.ac.cn). ERA5 data plotted in Supplementary Fig. 1 and Fig. 14 are available at https://cds.climate.copernicus.eu/. GPCP data plotted in Supplementary Figs. 1 and 15 are available at https://psl.noaa.gov/data/. MERRA-2 reanalysis data used in Supplementary Section 1.3.3 and plotted in Supplementary Fig. 20 are available at https://disc.gsfc.nasa.gov/datasets/. Further details of the datasets used in this study are given in Supplementary Section 6.

## Code availability

The WRF-Chem code can be downloaded from the official website: https://www2.acom.ucar.edu/wrf-chem. The InVEST (3.7.0) model is available at http://shichang-kang.sklcs.ac.cn/data-sharing.html. Data were analyzed with publicly available software: NCAR Command Language (NCL) and MATLAB. Major NCL scripts and WRF–Chem simulated results is deposited in http://shichang-kang.sklcs.ac.cn/data-sharing.html. Other scripts are available upon request. Shichang Kang (shichang.kang@lzb.ac.cn).

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

## Acknowledgements

We thank our participants for their time and attention. This study was funded by the Second Tibetan Plateau Scientific Expedition and Research Program (STEP)(2019QZKK0605), National Natural Science Foundation of China (42071096, 41901071), Strategic Priority Research Program of the Chinese Academy of Sciences, Pan-Third Pole Environment Study for a Green Silk Road (Pan-TPE) (XDA20040502), State Key Laboratory of Cryospheric Science (SKLCS-ZZ-2022), and the CAS "Light of West China" Program. We thank Rongjun Wang, Yuling Hu, and Jianwei Xu for their assistance in the paper revision.

## Author contributions

J.Y. and S.K. designed the research. J.Y. created all the figures. J.Y., D.C., and R.G. drafted the paper. R.G., D.C., L.Z., and H.D. performed analysis. J.Y., S.K., D.C., L.Z., Z.J., K.D., H.D., L.T., W.D., and R.G. contributed to the interpretation of the results. F.Y., M.R., and Y.L. commented on the manuscript.

## Competing interests

The authors declare no competing interests.
