## [Peer Review File · Nature Communications]

South Asian black carbon is threatening the water sustainability of the Asian Water TowerReviewer #1 (Remarks to the Author):

Please see attached review comments.

Reviewer #1 Attachment on the following page

Review comments to manuscript: NCOMMS-21-41910-T

South Asian black carbon threatening the water sustainability over the Asian Water Tower

by Yang et al.

General comments:

The manuscript seeks to investigate the relationship between the occurrence of long-range transport of black carbon (BC) from South Asia and changes in precipitation and glacial mass in the Tibetan Plateau and Himalayas. To answer this question, the authors integrate direct observations from a range of datasets with numerical simulations of WRF-Chem. While the topic investigated has important implications for the quantification of current and future water resources in Asia, the analyses presented are not sufficient to support the conclusions drawn. Major concerns about the methodology adopted, the analyses/results presented and novelty of the work have been identified, as highlighted in the specific and technical comments below, and need to be addressed.

Specific comments:

Language: The manuscript contains several typos, grammatical errors and unclear sentences. Proofreading from an English editor is needed after the text is fully revised. In the technical comments I include a non-exhaustive list of mistakes that need to be fixed.

Novelty of the work: The introduction does not clearly identify the existing research gap and therefore the relevance of the proposed research question. Only in the Discussion there is mention of discrepancies in results from prior modeling studies on the role of BC on Asian monsoons, but not enough details are provided to clearly identify the gap in knowledge that the study seeks to address.

Methods: the method section needs to be improved. If space is a constrain, more details should be provided in the SM. Below are some key issues pertaining the methodology adopted that have been identified.

- WRF-Chem simulations: Two WRF-Chem runs, one control run without BC emissions and one including BC, are performed and used to attribute changes in precipitation patterns and intensity due to the increase in anthropogenic emissions in the region. However, the following issues should be discussed/addressed.
 - 1) Is a resolution of 25 km enough to capture local convection? Prior works using the model to study monsoon patterns and extreme precipitation events in the region at similar resolution should be included to support the choice of this specific model setup.
 - 2) A more thorough model evaluation is needed. The only comparison between observations and simulations is presented in Figure 6 in the SM, which however only shows the two time series without any statistical metrics of model performance. Therefore, the sentence “a strong correlation...exists” included in the caption is not supported by any quantitative analysis. The Figure instead shows that a clear/systematic bias exists between observations and simulations, the latter ones overall overestimating observations. However, the bias is not mentioned nor quantified. More analyses are needed to discuss if it changes by season,

how it compares with the bias from other studies, and interpreted in light of the model setup/resolution adopted. As model simulations are used to identify drivers of precipitation changes, WRF-Chem output should be fully evaluated and results reported in the SM.

- Statistical analysis: the accumulative anomaly method should be at least briefly outlined. How was seasonality accounted for in the time series?

Data interpretation: The main conclusion of the manuscript is that BC has played a key role in dictating changes in the precipitation amount in the Tibetan Plateau. The authors identified an increasing trend in BC emissions since 1985 and suggested this may be the cause for the decreasing summer precipitation in the southern Tibetan Plateau since 2004 (as discussed at lines 99, 132-134 and 270). However, it is not clear how the attribution can be made as the positive trend in BC has started around 20 years before the one identified in precipitation. The association between BC role and precipitation is therefore weak and the physical/chemical mechanisms leading to such a delayed effect are not discussed/identified. At line 148, the authors then suggest a change in meteorology (meridional moisture and vertical wind components) has occurred since 2004 thus the role of these variables appears to be more strongly related to the trend change than the one of BC. The main conclusion of the manuscript is therefore not sufficiently supported by the analyses presented and more work is required to better understand the role of meteorology and emissions, and their interactions in dictating precipitation changes that are the focus of this study.

Technical comments:

Line 40-41: “in the Himalayas is” is repeated twice in the same sentence.

Line 58: a period is needed instead of comma after “Tibetan Plateau”.

Line 94: colon should be replaced with period after “decades”.

Line 101: “that” should be “what”.

Lines 130-132: this sentence is not grammatically correct and its content not clear.

Line 173: convective should be convection.

Line 172-175: this sentence needs to be rephrased as it is not clear.

Line 243: can you show the glacial retreat with your data analysis? As this is part of the key conclusions of the manuscript it should be supported by novel analysis and not just prior literature.

Line 286-289: this sentence does not make sense and needs to be reworded.

Line 357: a proper reference to the dataset used should be reported. What is the name of the dataset? It is fine to include a website address only if the data product used is clearly identified.

Line 561: Figure 1. The caption needs to be revised, the red polygon is in panel a and not b.

Reviewer #2 (Remarks to the Author):

By demonstrating covarying trends in precipitation and black carbon (BC) emissions over South Asia and modeled sensitivity of precipitation to local BC heating, this paper tries to establish a causal relationship between an increasing trend in BC emissions and a decreasing trend in glacier mass over southern Tibetan Plateau (TP). It is hypothesized that BC enhances convection and deplete water vapor within the Indian subcontinent, leading to a decline in water vapor transport to TP. The hypothesis and results are interesting and the claim is scientifically significant, but there are still gaps and concerns for a convincing mechanistic understanding of how the increases of BC emissions over South Asia drive a glacier mass decline in southern TP.

- 1) One gap is between BC emissions and atmospheric BC heating. The paper shows an increasing trend in BC emissions, which doesn't necessarily cause an increase in atmospheric BC concentration and heating. Atmospheric aerosols are largely removed during the Indian summer monsoon. Evidence is needed to show the trends in atmospheric BC loading and heating.**
- 2) The setup of model experiments is too idealized by simply turning on and off BC. With BC entirely removed from the sensitivity experiment, changes in precipitation are likely to be largely overestimated but BC still doesn't seem to have a substantial impact.**
- 3) The argument of moisture transport is largely speculative (L141-149). Fig. 2 does show a gradient in specific humidity and mean wind vectors, but it unnecessarily means the moisture convergence from South Asia to Southern TP has decreased during 2004-2018. A better way to quantify the moisture budget and convergence is needed.**
- 4) The last gap lies in the relationship between summer precipitation and glacier mass. Why does the study focus on the contribution of summer precipitation to glacier mass change? Could there be a seasonal to decadal shift in precipitation and its contribution to glacier mass accumulation? How much of glacier mass loss can be attributed to warming? These questions need to be answered to fill in the gap.**

A few other comments for authors to consider:

- 1) The statement in L39-41 of the abstract is misleading. The decreases in glacier mass are not directly attributable to BC emissions in South Asia.**
- 2) It is argued that changes in convective precipitation is minimal, but CAPE has a significant increase, which is seemingly contradictory.**
- 3) L197-199: According to aerosol indirect effect, more CCN would lead to less precipitation. It's unclear what argument this result tries to support.**
- 4) L74-75: this statement needs more context. The impact of black carbon on moisture convergence appears to depend on the vertical location of the heating layer and the spatiotemporal scales.**
- 5) Figure 1: what is accumulative anomaly? Please define it and justify the use of this quantity instead of the simple anomaly of summer mean. More detailed information of each figure is expected to be described in the caption. Is the BC emission from all sources? Is it also summer mean anomaly? It is surprising to see the smooth trends (lack of interannual variability). Why didn't the WRF simulations use the same BC emissions (instead of INTEX-B)? The trends of BC emissions over South Asia look different in Supplementary Fig. 9.**
- 6) Fig. 4: What does the CCN change represent? Supersaturation? At which height? It's quite surprising to see such a big change in CCN due to BC emissions alone.**
- 7) Supplementary Fig. 6: The WRF model seems to simulate a decreasing summer precipitation from 2007 to 2015 over South Asia, which is opposite to CRU data.**

Editor (Remarks to the Author)

Thank you again for submitting your manuscript "South Asian black carbon threatening the water sustainability over the Asian Water Tower" to Nature Communications. We have now received reports from 2 reviewers and, after careful consideration, we have decided to invite a major revision of the manuscript.

As you will see from the reports copied below, the reviewers raise important concerns. We find that these concerns limit the strength of the study, and therefore we ask you to address them with additional work. Without substantial revisions, we will be unlikely to send the paper back to review. In particular, the referees raise concerns about the methods, analytical choices, and mechanistic explanations for the relationship between black carbon and water vapor transport to the Tibetan Plateau. To move forward, more detailed case for the novelty of results and methods, a more in-depth model evaluation, and support for main conclusions as highlighted in the review reports must be thoroughly addressed within the main text.

If you feel that you are able to comprehensively address the reviewers' concerns, please provide a point-by-point response to these comments along with your revision. Please show all changes in the manuscript text file with track changes or colour highlighting. If you are unable to address specific reviewer requests or find any points invalid, please explain why in the point-by-point response.

Response: Thank you very much for the invitation to further revise this manuscript. We have fully addressed the concerns raised by reviewers, including but not limited to the following:

(1) We now present the methods in detail in Supplementary Section 1, including the accumulative anomaly method, WRF-Chem model, and a monthly-scale glacial mass balance model and so on.

(2) As for the analytical choices, by using the lag correlation analysis, we found that there is an insignificant lag relationship between summer precipitation over the Tibetan Plateau with the South Asian black carbon (Supplementary Section 4.1). Thus, we decided to focus on the contemporaneous impact of South Asian black carbon in this study.

(3) To provide better mechanistic explanations for the relationship between black carbon and water vapor transport to the Tibetan Plateau, we calculated the moisture flux and its divergence from 1989 to 2018 by using a the reanalysis dataset (Supplementary Section 2.2). Results show a decreased trend of incoming water vapor over the southern Tibetan Plateau, but a strengthening of water vapor convergence over South Asia in recent years. Based on simulated results, we have added the calculation of South Asian black carbon affecting the divergence of moisture flux in Fig. 4b of the revised manuscript, and substantiate its effect on water vapor transport to the Tibetan Plateau by enhancing moisture convergence over the South Asia.

In addition, we analyzed the cause of the strengthening of the cyclonic circulation in the eastern Indian subcontinent due to black carbon (Lines 197-206 in revised manuscript), which weakens the northward transport of moisture to the southern Tibetan plateau.

(4) To illustrate the novelty of the work, we have identified the gaps between previous works and this study in the third paragraph of the Introduction section and discussed our findings in the Discussion section in the revised manuscript.

(5) A more in-depth model evaluation has been conducted, including the comparison of performances of the model with different physical scheme combinations, evaluating the WRF-Chem performance for precipitation and black carbon based on in situ observations and reanalysis datasets, model uncertainty analysis, and discussing the bias of this study in relation to other studies. See Supplementary Section 1.3.

(6) We have added more analysis to support the key conclusions of the manuscript. For example, for glacial retreat over the Tibetan Plateau, we reconstructed the time series of reference-surface mass balance for 1979–2014 periods using an empirical model, estimated contributions of (summer) precipitation and atmospheric warming to glacier mass change, as well as spatial heterogeneity. See Supplementary Section 5.

Reviewer #1 (Remarks to the Author):

Please see attached my review comments.

Reviewer #1 Attachment on the following page

Review comments to manuscript: NCOMMS-21-41910-T

South Asian black carbon is threatening the water sustainability of the Asian Water Tower by Yang et al.

General comments:

The quality of the manuscript has improved compared to the first submission and the authors have address most of my initial comments. However major concerns about the methodology adopted, the analyses/results presented, as well as the presentation of the results remain. Below I outline some specific and technical comments that need to be addressed.

Specific comments:

Language: While the writing of the manuscript has improved, several typos, grammatical and punctuation errors, as well as unclear sentences still remain. I include a few examples in the technical comments, but a much deeper revision of the entire manuscript (and supplementary materials) is needed. In general, most of the figure captions are inadequate as they only provide partial or inaccurate information about what it is displayed. A full revision of the captions to make the figure self-explanatory is needed.

Reproducibility of results: Most of the results are not reproducible as key details are missing. This is not acceptable for a scientific publication. I pointed out to key issues, but a full revision of the main manuscript and supplementary materials should be done. The results presented in the figures are not clearly described either in the captions or the main manuscript. Some, not exhaustive, examples are provided in the technical comments.

Interpretation of the results: more depth and coherence in the interpretation of the results should be included. This for example pertains the fact that different years/time periods are used for different analyses, without providing a clear justification for them. Discussion of the spatial patterns of the significance (or not significance) of the changes in precipitation and proper attribution of such changes should be better addressed.

Technical comments:

Line 68: “Documentation reveals”, please rephrase as not suitable for a scientific publication.

Line 78: “effected” should be “affected”.

Line 95: “:” should be “.”

Line 135: “transportation” should be “transport”. This error occurs at other instances through the manuscript.

Line 142: remove “the” from “the opposite”.

Line 163: Please explain what is the purpose of the lag correlation analysis.

Line 168: it’s not clear what do you mean by “over the southern aspect”.

Line 211: “tends” should be “tend”.

Line 248: remove “has resulted in” as it is a repetition of the same verb at the beginning of the sentence.

Line 268: “this accompanied...” needs to be rephrased/clarified.

Line 270-273: please rephrase.

Line 274: “further” is repeated twice, please fix it.

Line 289: you use “:” when instead it should be a “.” or fix the capital letters afterward. This error occurs at several instances in the manuscript.

Line 301: “black carbons” should be “black carbon’s”.

Line 317: this paragraph is not related to your results, so you should explicitly clarify that.

Line 334: missing space after “Tibetan plateau”.

Line 349: more details are needed about the emission dataset which is not even mentioned either in the WRF-Chem setup in the supplementary materials or in the main text at page 12-13. Did the anthropogenic emissions vary by year? Could you show the trends in BC for the region of interest as they are quite critical to what you are testing? Listing a website in the main manuscript is not appropriate, a proper reference should be included and the website link provided in the data availability section. Further, the website does not work, so I could not verify any information about the emissions.

Line 381: this sentence needs to be rephrased to clarify how you perturbed BC emissions and where/when.

Line 386: you need to justify why only those months. Couldn’t there be long term impacts on circulation that you are not able to see by focusing only on those months?

Figures

As mentioned in the first review, the figures captions are not appropriate. They lack fundamental details that the reader would need to be able to interpret them and to reproduce the results. Below are some examples.

Figure 1: It is not mentioned which dataset is used to produce those maps and time series. For the time series it is also not clear which location they refer to. Are you showing a spatial average or are you focusing on a specific location? Not knowing if you are displaying model results or direct observations it is hard to infer. It’s not stated what the p-value reported refers to. Is it on the slope? Why are there two lines for each color (in panel c). It is not stated the reference period used to compute the accumulative anomaly.

Figure 2: Here and in other part of the text there is discrepancy among the years analyzed. Most of the analyses refer to 2007-2016, but here you mention different time frames. The period of interest should be consistent with your key analyses to provide a coherent interpretation of your results. A deeper interpretation of the signal shown in panels c and d should be provided. What is the role of the different time periods analyzed in the significance and direction of the correlation?

Figure 3: how is the “large scale precipitation change” shown in panel b defined? This result is not clear and not reproducible.

Figure 4: the titles are spelled wrong “moistureflux”. Also it is not clear how you define/compute the change. This result is also not reproducible.

Supplementary Materials

WRF-Chem setup: please add the spatial and temporal resolution of the different emission datasets used and add the anthropogenic emissions.

Figure S3: the summer precipitation mentioned in the caption is not shown.

Figure S11: how do you explain the systematic low bias in the simulations? How is this impacting your results and conclusions? This should be discussed in the main manuscript as well.

Reviewer #2 (Remarks to the Author):

I appreciate the authors' effort in addressing all of my previous comments. The additional analyses and supplementary figures are very helpful. One remaining concern is the Figure 4d. Although the large change in CCN number concentrations due to black carbon emissions has been explained, I don't think it's appropriate to simply use the sum of number concentrations of all model layers because of the strong dependence on the total number of model layers. Either mass-weighted average or column-integrated number concentration (in units of $\#/m^2$) can be used instead.

Replies to the reviewers' comments follow:

Reviewer #1

General comments:

The quality of the manuscript has improved compared to the first submission and the authors have address most of my initial comments. However major concerns about the methodology adopted, the analyses/results presented, as well as the presentation of the results remain. Below I outline some specific and technical comments that need to be addressed.

Response: We thank the reviewer for the recognition of our improvement submission, as well as further careful and helpful suggestions. We have fully addressed the major concerns in the revised manuscript that were highlighted by the reviewer, including but not limited to (1) grammatical and punctuation errors as well unclear sentences; these have been corrected, (2) the WRF-Chem configuration is now improved re: explanation and justification, (3) results are better presented in figure captions and/or in the manuscript, (4) different periods used for different analyses have been justified and, (5) a discussion of the spatial patterns of the significance (or not) of the changes in precipitation and a proper attribution of such changes is addressed more fully. Specific responses and revisions are presented as follows.

Specific comments:

Language: While the writing of the manuscript has improved, several typos, grammatical and punctuation errors, as well as unclear sentences still remain. I include a few examples in the technical comments, but a much deeper revision of the entire manuscript (and supplementary materials) is needed. In general, most of the figure captions are inadequate as they only provide partial or inaccurate information about what it is displayed. A full revision of the captions to make the figure self-explanatory is needed.

Response: Thank you very much for your careful check. We have corrected these errors and revised unclear sentences (see Table 2 here). In addition, we tried our best to revise fully the entire manuscript and the supplementary materials marking corrections or additions in red. We have revised all the figure captions as requested.

Reproducibility of results: Most of the results are not reproducible as key details are missing. This is not acceptable for a scientific publication. I pointed out to key issues, but a full revision of the main manuscript and supplementary materials should be done. The results presented in the figures are not clearly described either in the captions or the main manuscript. Some, not exhaustive, examples are provided in the technical comments.

Response: We apologize for the missing details and thanks a lot for your helpful and valuable suggestions. We have now conducted a deeper revision of the figure captions in the manuscript and supplementary materials. We have also improved the results presented in the manuscript, including but not limited to the examples raised in the technical comments.

Interpretation of the results: more depth and coherence in the interpretation of the results should be included. This for example pertains the fact that different years/time periods are used for different analyses, without providing a clear justification for them. Discussion of the spatial patterns of the significance (or not significance) of the changes in precipitation and proper attribution of such changes should be better addressed.

Response: Following your suggestions, we have now included more depth in the interpretation of the results. Yes, it is a fact that different years/time periods are used which pertain to the analysis at hand. We will provide a justification for them.

First, there exists matters of availability among various datasets. In Fig. 1 in the revised manuscript and Supplementary Fig. 28, the APHRODITE precipitation dataset ends at 2015, thus we are limited by the end-date. In Fig. 2 in the manuscript, the black carbon emissions from Peking University only covers the period 1960 to 2014 and so, we can only run the correlation analysis up to 2014, as well as analyzing the South Asian area-averaged black carbon emissions for the period of 1961–2014. In Supplementary Fig. 15, the GPCP precipitation starts at 1979, so we used this dataset's complete record that spanned 1979–2016.

We selected different years/time periods for different analyses as was necessary to identify trends, i.e., you use the dataset that affords you the greatest timeline. For the initial analysis, Fig. 1 in the manuscript, Supplementary Fig. 15, and Supplementary Fig. 16 identified a decreasing trend in summer precipitation in the southern Tibetan plateau that started in 2004; this was based on both the CRU dataset (1961–2016, Fig. 1b in the manuscript) and the GPCP dataset (1979–2016, Supplementary Fig. 15), and was further confirmed by in-situ observations (2004–2016, Supplementary Fig. 16a) along with the APHRODITE dataset (2004–2015, Fig. 1c in the manuscript).

Next, we selected the 2004 as a time node/inflexion point to investigate the annual change of summer meridional moisture transport from South Asia to the southern Tibetan plateau, where we compared the meridional moisture transport difference between 1989–2003 and 2004–2018 using the ERA-Interim reanalysis dataset, i.e., fifteen years before and after the inflexion point of 2004. These results are shown in Fig. 2a and Fig. 2b in the manuscript, and Supplementary Fig. 17, which show a decrease in incoming moisture over the southern Tibetan Plateau accompanied with strengthened moisture convergence over South Asia. Noting from the literature that black carbon can induce atmospheric moisture convergent conditions, we plotted and remarked upon the correlation between area-averaged precipitation over the southern Tibetan plateau and the black carbon emissions over the South Asia since the 2004 inflexion point, in addition to a comparison that included the complete dataset record that started in 1961.

Finally, we conducted WRF-Chem simulations for summer months for ten-years (2007–2016) after 2004; this was to investigate whether and how atmospheric black carbon over south Asia might affect and cause diminishing summer precipitation over the southern Tibetan plateau, as well as its contribution towards glacial mass decline. Prior WRF-Chem simulations (Kumar et al., 2015, Xu et al., 2018; Yang et al., 2018; Chen et al., 2018; Singh et al., 2020) represented well the black carbon concentrations over South Asia and the Tibetan Plateau; these studies were driven by MOZART chemical output starting in 2006. Given the 2006 start date of the MOZART real-time data, it was reasonable to select the ensuing period of 2007–2016 to examine black

carbon's role in precipitation change and glacial decline.

In Table 1 here (Supplementary Table 8), we have itemized the datasets selected periods, the analysis undertaken and, a brief explanation. We have added above-mentioned information for justification of different years/time periods used for different analyses in Supplementary Section 6.

Table 1 Summary of iterative analyses and corresponding dataset periods used in this study.

Different analyses	Datasets	Note
I: Identification of a decreasing trend in summer precipitation in the STP since 2004.	CRU (1961–2016, Fig. 1b)	
	GPCP (1979–2016, Fig. S15)	This dataset is available since 1979
II: Confirmation of decreased summer precipitation in STP since 2004.	In situ observations (2004–2016, Fig. S16a)	
	APHRODITE (2004–2015, Fig. 1c)	This dataset is only updated to 2015
III: Comparing the change of incoming moisture to the TP before and after 2004, indicating a decrease in incoming moisture since 2004.	ERA-Interim (1989–2018, Fig.2a, Fig. 2b, and Fig. S17)	To reveal the averaged annual change for fifteen years before as well as after 2004.
IV: Discussion of the correlation coefficient between summer precipitation decreased over the STP and BC emissions over the South Asia since 2004.	Peking BC emission (1961–2014, Fig. 2c and Fig. 2d)	This dataset period is only covered from 1960 to 2014.
	CRU precipitation (1961–2014, Fig. 2c and Fig. 2d)	
V: Revealing the annual averaged effect of Asian BC on summer precipitation decrease over the STP, and its contribution to glacial mass loss.	WRF-Chem simulations (2007–2016, Fig. 3 and Fig.4)	Prior similar simulations using the MOZART represented well the BC after 2006, so 2007–2016 selected.
	InVEST evaluation (2007–2016, Fig. 5)	

*BC and STP are the abbreviations of black carbon and southern Tibetan Plateau, respectively.

In addition, we have provided a more in-depth interpretation of Fig. 2c and Fig. 2d in Lines 166–168 in the manuscript. Specifically, the different time periods in both significance and correlation coefficient suggest a strong relationship between increased south Asian black carbon (Supplementary Fig. 19) and the decrease of summer precipitation over the southern Tibetan plateau since the 21st century (Fig. 2c in the manuscript), compared to the correlation between the two is insignificant for the longer period of 1961–2014 (Fig. 2d in the manuscript).

References:

- Chen, X. T., Kang, S. C., Cong, Z. Y., et al. Concentration, temporal variation, and sources of black carbon in the Mt. Everest region retrieved by real-time observation and simulation. *Atmos. Chem. Phys.* **18**, 12859–12875 (2018).
- Dong, W. et al. Summer rainfall over the southwestern Tibetan Plateau controlled by deep convection over the Indian subcontinent. *Nat. Commun.* **7**, 10925(2016).
- Kumar, R., Barth, M. C., Pfister, G. G., et al. What controls the seasonal cycle of black carbon aerosols in India? *J. Geophys. Res.-Atmos.* **120**, 7788–7812 (2015).
- Singh, P., Sarawade, P., Adhikary, B. Transport of black carbon from planetary boundary layer to free troposphere during the summer monsoon over South Asia. *Atmos. Res.* **235**, 104761 (2020).
- Xu, R. G., Tie, X. X., Li, G. H., Zhao, S. Y., Cao, J. J., Feng, T., Long, X. Effect of biomass burning on black carbon (BC) in South Asia and Tibetan Plateau: The analysis of WRF-Chem modeling. *Sci. Total. Environ.* **645**, 901–912 (2018).
- Yang, J. H., Kang, S. C., Ji, Z. M. Sensitivity Analysis of Chemical Mechanisms in the WRF-Chem Model in Reconstructing Aerosol Concentrations and Optical Properties in the Tibetan Plateau. *Aerosol. Air. Qual. Res.* **18**, 505–521 (2018).

Technical comments:

Line 68: “Documentation reveals”, please rephrase as not suitable for a scientific publication. Line 78: “effected” should be “affected”. Line 95: “:” should be “.”

Line 135: “transportation” should be “transport”. This error occurs at other instances through the manuscript. Line 142: remove “the” from “the opposite”.

Line 211: “tends” should be “tend”. Line 248: remove “has resulted in” as it is a repetition of the same verb at the beginning of the Sentence. Line 270–273:

please rephrase. Line 274: “further” is repeated twice, please fix it. Line 289: you use “:” when instead it should be a “.” or fix the capital letters afterward. This error occurs at several instances in the manuscript. Line 301: “black carbons” should be “black carbon’s”. Line 334: missing space after “Tibetan plateau”.

Response: Thank you very much for your careful check. We have corrected these errors as noted in Table 2. In addition, we invited a native English speaker to help us check the updated manuscript again.

Table 2 Technical comments on language and our responses

Comments	Responses
Line 68: “Documentation reveals”, please rephrase as not suitable for a scientific publication.	Thank you so much for your careful check. We have rephrased this sentence in Lines 69–70 of the revised manuscript.
Line 78: “effected” should be “affected”.	We have corrected this typo error in Line 79 .
Line 95: “:” should be “.”	We have replaced “:” with the “.” in Line 97 .
Line 135: “transportation” should be “transport”. This error occurs at other instances through the manuscript.	We have corrected this error through the manuscript.
Line 142: remove “the” from “the opposite”.	We have revised it in Line 146 .
Line 211: “tends” should be “tend”.	We have revised the sentence in Line 217 .
Line 248: remove “has resulted in” as it is a repetition of the same verb at the beginning of the Sentence.	We have removed “has resulted in” in Line 255 .
Line 270–273: please rephrase.	We have rephrased this sentence in Lines 275–281 .
Line 274: “further” is repeated twice, please fix it.	We have corrected this error in Line 282 .
Line 289: you use “:” when instead it should be a “.” or fix the capital letters afterward. This error occurs at several instances in the manuscript.	We have corrected this punctuation error through the manuscript.
Line 301: “black carbons” should be “black carbon’s”.	We have replaced “black carbons” with “black carbon’s” in Lines 310 .
Line 334: missing space after “Tibetan plateau”.	We have corrected this error in Line 345 .

Line 163: Please explain what is the purpose of the lag correlation analysis.

Response: The purpose of the lag correlation analysis was to identify if there was a delayed effect of increased South Asian black carbon (BC) on reduced summer precipitation over the southern Tibetan Plateau. As pointed out in your initial comments, an ever-increasing trend in BC emissions since 1985 may be the cause for the decreasing summer precipitation in the southern Tibetan Plateau since 2004. We have found no lag relationship between the two, and have clarified such in Lines 168–171 in the manuscript.

Line 168: it's not clear what do you mean by “over the southern aspect”.

Response: Thanks for your careful reading. We want to examine the cause and effect of atmospheric black carbon over south Asia on summer precipitation over the southern Tibetan plateau. We have reworded it in Lines 172–173 in the manuscript.

Line 268: “this accompanied...” needs to be rephrased/clarified

Response: We have rephrased this sentence in Lines 275–277 in the manuscript.

Line 317: this paragraph is not related to your results, so you should explicitly clarify that

Response: Following your valuable suggestion, we have clarified this sentence in Lines 325–327 in the manuscript.

Line 349: more details are needed about the emission dataset which is not even mentioned either in the WRF-Chem setup in the supplementary materials or in the main text at page 12–13. Did the anthropogenic emissions vary by year? Could you show the trends in BC for the region of interest as they are quite critical to what you are testing? Listing a website in the main manuscript is not appropriate, a proper reference should be included and the website link provided

in the data availability section. Further, the website does not work, so I could not verify any information about the emissions.

Response: Many thanks for this comment. The different emission datasets used to drive the WRF-Chem is now introduced in Lines 374–383 in the “WRF-Chem model simulations” section in the manuscript. We have now summarized in more detail the emission datasets used in the WRF-Chem simulations as well as the differences between the control and sensitivity experiments in Table 3 here (Supplementary Table 2). The default anthropogenic emissions from INTEX-B are only available for 2006 at a horizontal resolution of $1^\circ \times 1^\circ$; this was replaced by the real-time anthropogenic emissions from MOZART at 6-hourly temporal resolution. The MOZART emissions used in this study varies by year.

Table 3 Summary of different emission datasets used in WRF-Chem simulations as well as their changed in different experiment.

Emission datasets	Resolutions		Changes in experiments	
	Spatial	Temporal	Control	Sensitivity
MEGAN biogenic emissions	0.03°	Month	Original values	Set to zero over South Asia
FINN fire emissions	500m	Hour	Original values	Set to zero over South Asia
MOZART used for anthropogenic emissions	1°	6-hourly	Original values	Set to zero over South Asia

* South Asia is represented by the red polygon in Fig. 1a in the revised manuscript.

By analyzing the latest version of the MERRA-2 dataset (Modern-Era Retrospective analysis for Research and Applications version 2), we found that atmospheric black carbon concentrations were ever-increasing over time (Fig. 1 here), i.e., for the length of the dataset from 1986 to 2016. In addition, we calculated the summer monthly black carbon emissions over South Asia since 1960 in Supplementary Fig. 19 and, also found an increasing trend.

Fig. 1 Summer black carbon concentration over the period 1986 to 2016 as derived from MERRA-2.

Following your careful suggestion, we have removed and moved website links in the manuscript to the data availability section in the manuscript or the Supplementary Section 6. Moreover, we have now summarized more fundamental details of the datasets used in this study in Table 4 here (Supplementary Table 7), including the data type, spatial and temporal resolutions, and the time/period used. We have further checked all the websites as to access and confirm this is the case. The website link for black carbon emission inventories from Peking University (<http://inventory.pku.edu.cn/>) is sometimes intermitted but, as of now seems to be stable and accessible. If anyone has difficulty in acquiring the data from the Peking University, they can send us a note/ request and we can upload the data onto our website (we have uploaded the annual mean black carbon emissions from Peking University for the period 2001 – 2014 on our website).

Table 4 More details of the datasets used in this study

Parameter	Dataset name	Type	Resolutions (spatial, temporal)	Period
Precipitation	CRU	Gridded	0.5°×0.5°, month	1961–2016
	APHRODITE	Gridded	0.25°×0.25°, day	2004–2015
	ERA5	Gridded	0.25°×0.25°, month	2016
	GPCP	Gridded	2.5°×2.5°, month	1979–2016
	CMDSC	In situ	China, day	2004–2016
	UCC	In situ	South Asia, day	2016
Specific moisture	ERA-Interim	Gridded	0.25°×0.25°, month	1989–2018
Wind	ERA-Interim	Gridded	0.25°×0.25°, month	1989–2018
Moisture flux	ERA-Interim	Gridded	0.25°×0.25°, month	1989–2018
Divergence of moisture flux	ERA-Interim	Gridded	0.25°×0.25°, month	1989–2018
Black carbon concentration and wet deposition	MERRA-2	Gridded	0.5 ° × 0.625 °, month	1986, 2001, 2016
Black carbon concentration	APCC	In situ	South Asia and TP, day	2016
Black carbon emission	Peking University	Gridded	0.1°×0.1°, month	1961–2014

*UCC data is derived from the Utah Climate Center (<https://climate.usu.edu/>).

Line 381: this sentence needs to be rephrased to clarify how you perturbed BC emissions and where/when.

Response: This is good suggestion. We have rephrased the third paragraph in Lines 398–403 in the “WRF-Chem model simulations” section in the manuscript. To summarize, in the control simulations the initial black carbon and emissions for black carbon was unchanged, including biogenic emissions, fire emissions, and anthropogenic emissions. Whereas in the sensitivity simulation, the initial black carbon and all the emissions of black carbon were set to zero over South Asia (the red polygon in Fig. 1a in the revised manuscript) at each time.

Line 386: you need to justify why only those months. Couldn't there be long term impacts on circulation that you are not able to see by focusing only on those months?

Response: First, the selection of June to September as summer months (to analyze the precipitation change over the Tibetan Plateau) was based on lots of prior studies (Dong et al., 2016; Singh et al., 2014; Zhang et al., 2019); these studies found summer precipitation accounts for more than 60% of the total precipitation over the Tibetan Plateau. In this study, we were focused on the effects of South Asian black carbon on precipitation over the southern Tibetan Plateau in the summer months averaged over the period 2007 to 2016.

In addition, black carbon has a short atmospheric lifetime and so, any climate forcing/effect is transitory as it can be removed by atmosphere within a few days to weeks depending on precipitation events and surface contact (Bond et al., 2012). Therefore, it did not seem practical to consider long-term impacts given the nature of our study. As a final note, we used one month as the model spin-up time to reach a state of statistical equilibrium.

We would also like to underscore that the comparisons we implemented with all available observational data were in good agreement when it came to reproducing black carbon and, summer precipitation both spatially and in magnitude (Supplementary Section 1.3). Thus, the simulations were conducted only for those several months during the ten-year period (in modeling the average effects of South Asian black carbon on summer precipitation over the southern Tibetan Plateau) where verification was possible.

References:

- Dong, W. et al. Summer rainfall over the southwestern Tibetan Plateau controlled by deep convection over the Indian subcontinent. *Nat. Commun.* **7**, 10925(2016).
- Singh, D., Tsiang, M., Rajaratnam, B. & Diffenbaugh, N. S. Observed changes in extreme wet and dry spells during the South Asian summer monsoon season. *Nat. Clim. Change.* **4**, 456–461 (2014).

Zhang, C., Tang, Q., Chen, D., van der Ent, R. J., Liu, X., Li, W., & Haile, G. G. Moisture source changes contributed to different precipitation changes over the northern and southern Tibetan Plateau. *J. hydrometeorol.* **20**, 217–229 (2019).

Bond, T.C., et al. Bounding the role of black carbon in the climate system: a scientific assessment. *J. Geophys. Res. Atmos.* **118**, 5380–5552 (2013).

Figures

As mentioned in the first review, the figures captions are not appropriate. They lack fundamental details that the reader would need to be able to interpret them and to reproduce the results. Below are some examples.

Response: We are sorry for not providing appropriate captions for figures as mentioned in the first review. Following your careful and valuable suggestions, we have updated to help a reader understand and correctly interpret and, if desired, reproduce the figures; this was done for both the main manuscript and supplementary materials.

Figure 1: It is not mentioned which dataset is used to produce those maps and time series. For the time series it is also not clear which location they refer to. Are you showing a spatial average or are you focusing on a specific location? Not knowing if you are displaying model results or direct observations it is hard to infer. It's not stated what the p-value reported refers to. Is it on the slope? Why are there two lines for each color (in panel c). It is not stated the reference period used to compute the accumulative anomaly.

Response: Thanks for your careful comments. We have supplemented all the datasets used in Fig. 1 in the manuscript, including summer precipitation using the CRU dataset (2001–2016), summer 500 hPa wind fields using the ERA-Interim dataset (2001–2016) and, black carbon using the Peking University's emissions datasets (1961–2014). (b) and (c) are a spatial mean. We have added “area-averaged” in the caption.

Yes, the p-value in (c) refers to the slope. The three dotted lines in (c) represent the

trends of summer precipitation during different periods, i.e., the longest is for 1961–2016, the second longest is for 1961–2003, and the shortest is for 2004–2016. The reference period used to compute the accumulative anomaly in (b) is consistent with that in (c). We have added scales at the bottom of (b). The caption in Fig.1 should now be self-evident.

Figure 2: Here and in other part of the text there is discrepancy among the years analyzed. Most of the analyses refer to 2007–2016, but here you mention different time frames. The period of interest should be consistent with your key analyses to provide a coherent interpretation of your results. A deeper interpretation of the signal shown in panels c and d should be provided. What is the role of the different time periods analyzed in the significance and direction of the correlation?

Response: Yes, there is a discrepancy among the years analyzed. To elucidate, in Fig.2b in the manuscript, we compared the meridional moisture transport change fifteen years before and fifteen years after 2004, i.e., 1989–2003 and 2004–2018; the rationale being, and following on from Fig. 1 in the manuscript where we identified a decreasing trend in summer precipitation in the southern Tibetan Plateau that started in 2004; this was corroborated by ensuing analysis, i.e., regulated by long-range moisture transport.

We have provided a deeper interpretation of Fig. 2c and Fig. 2d in Lines 166–168 in the revised manuscript and, especially in the caption. The separated timespans around the inflexion point of 2004 are key in that they infer through statistical significance and direction of the relationship as indicated by the sign of the coefficient that increased south Asian black carbon over time plays an important role in the decrease of summer precipitation over the southern Tibetan plateau after 2004 (Fig. 2c in the manuscript) (also, please refer to Supplementary Fig. 19). Of course, the correlation analysis is simply a statistical measure of the relationship that lead to formulation of a hypothesis – hence the investigative modeling and climate diagnostic analysis that followed to determine possible causation.

Figure 3: how is the “large scale precipitation change” shown in panel b defined? This result is not clear and not reproducible.

Response: Large scale and convective precipitation are separate output variables in WRF-Chem so easily reproducible.

As to the definition, the “large-scale precipitation” is the accumulated total grid scale precipitation, driven by large-scale atmospheric dynamics. The “large-scale precipitation change” is defined as the large-scale precipitation change caused by South Asian black carbon, which was calculated based on WRF-Chem simulations, i.e., the values of large-scale precipitation in the control simulations (with effects of South Asian black carbon) minus that in the insensitivity simulations (without effects of South Asian black carbon). We have supplemented these details in the figure caption.

Figure 4: the titles are spelled wrong “moistureflux”. Also it is not clear how you define/compute the change. This result is also not reproducible.

Response: We have corrected the “moistureflux”. The change in the various variables due to South Asian black carbon (ref., Fig. 4 in the manuscript) in the revised manuscript were calculated using WRF-Chem output, i.e., the values of variables in the control simulations (with effects of South Asian black carbon) minus that in the sensitivity simulations (without effects of South Asian black carbon). We have supplemented Fig.4's caption to clarify these.

Supplementary Materials

WRF-Chem setup: please add the spatial and temporal resolution of the different emission datasets used and add the anthropogenic emissions.

Response: Following the reviewer’s suggestion, we have summarized the different emission datasets used in WRF-Chem simulations as well as the initial prescribed conditions of the control and sensitivity experiments (please refer to Table 3 here) (Supplementary Table 2). We have added further details in the “WRF-Chem model

simulations” section in the main text. The default use of anthropogenic emission was obtained from the INTEX-B (only available for year 2006), which was replaced by the real-time MOZART output in this work. Thus, we only introduce the MOZART as the anthropogenic emission source in Supplementary Table 2.

Figure S3: the summer precipitation mentioned in the caption is not shown.

Response: The black dots refer to the location of in-situ observations for summer precipitation. We have now clarified it in the caption.

Figure S11: how do you explain the systematic low bias in the simulations? How is this impacting your results and conclusions? This should be discussed in the main manuscript as well.

Response: Thanks for your careful reading. In Supplementary Fig. 11, we evaluated the WRF-Chem performance for four pollution events at QOMS. The low simulation bias arises at this site, at least in part, because the model grid is a point source regional average computation over 25km, whereas the observation site is more strongly influenced by complex local topography that is not represented well in the model because of the model’s coarse resolution.

However, we note that for all the observation sites in the study area, the model did not consistently underestimate the black carbon concentration, such as at such as at Kathmandu and Lanzhou in Supplementary Fig. 8. Moreover, the model generally represented the black carbon concentration at these APCC stations well (Supplementary Fig. 9). Therefore, it seems that there is no systematic low bias in this particular WRF-Chem simulation setup for black carbon simulation.

Furthermore, as shown in Supplementary Section 1.3.3, when compared with the available in-situ observations and gridded dataset, the model does reproduce satisfactory results of seasonal black carbon concentrations as well as capturing its spatial variability and magnitudes, albeit low bias occurred at the QOMS site (ref., Supplementary Fig. 11) where complex terrain was the case. Therefore, we felt

confident in the model setup. We have also provided supplementary text (Lines 388–394) in the revised manuscript.

Reviewer #2 (Remarks to the Author)

I appreciate the authors' effort in addressing all of my previous comments. The additional analyses and supplementary figures are very helpful. One remaining concern is the Figure 4d. Although the large change in CCN number concentrations due to black carbon emissions has been explained, I don't think it's appropriate to simply use the sum of number concentrations of all model layers because of the strong dependence on the total number of model layers. Either mass-weighted average or column-integrated number concentration (in units of $\#/m^2$) can be used instead.

Response: We thank very much for your endorsement of our effort to improve our work as well as your most helpful suggestions. We fully accept that it's inappropriate to simply use the sum of CCN number concentrations at all model layers. Following your careful suggestion, we have now calculated the weighted vertical average of CCN change due to black carbon emissions, as show in Fig. 2 below. We have replaced Fig. 4d with this figure in the revised manuscript. In addition, the corresponding code and results are now available at <http://shichang-kang.skics.ac.cn/data-sharing.html>.

Many thanks once more for your time and meaningful contribution to improve our work.

Fig. 2 South Asian black carbon triggered the change of weighted vertical average number concentration of cloud condensation nuclei (CCN) during 2007–2016.

Reviewer #1 (Remarks to the Author):

The manuscript has greatly improved and the authors have addressed all my major and detailed comments. However I would like to point out a couple of minor edits needed in the supplementary materials:

Table S4: It's not clear/specified which variable has been used for the model evaluation. While in the text there is some information, an explicit mention is needed in the caption. Also, what are the units of the statistical metrics computed?

Figure S5: similarly to Table S4, it's not clear what variable is evaluated. Please specify.

Figure S13: It's not clear what is displayed on the x-axis. If time, please specify it and it's units. Also why the x-axis is different from panel 1 and b?

Replies to the reviewers' comments follow:

Reviewer #1 (Remarks to the Author)

The manuscript has greatly improved and the authors have addressed all my major and detailed comments. However, I would like to point out a couple of minor edits needed in the supplementary materials.

Response: We thank very much for your endorsement of our effort to improve our work as well as your most helpful suggestions. We have revised a couple of minor edits in the supplementary materials according your suggestions. Specific responses and revisions are presented as follows.

Table S4: It's not clear/specified which variable has been used for the model evaluation. While in the text there is some information, an explicit mention is needed in the caption. Also, what are the units of the statistical metrics computed?

Response: Thanks for your careful reading. In Table S4, we evaluated the model performance in summer precipitation, thus have specified it in the caption. Moreover, we have added the units of the statistical metrics in Table S4, except for the TMP which does not have unit.

Figure S5: similarly to Table S4, it's not clear what variable is evaluated. Please specify.

Response: In Table S4, the summer precipitation is evaluated. We have specified it in the caption.

Figure S13: It's not clear what is displayed on the x-axis. If time, please specify it and it's units. Also why the x-axis is different from panel 1 and b?

Response: In Figure S13, the x-axis show the grid number of different WRF-Chem simulations. There is different spatial resolutions among different WRF-Chem configurations (i.e., 25 km and 8 km), so the x-axis (grid number) is different between panel 1 and b.

Many thanks once more for your time and careful reading to improve our work.